# Different Nitrogen Consumption Patterns in Low Temperature Fermentations in the Wine Yeast *Saccharomyces cerevisiae*

**DOI:** 10.3390/foods13162522

**Published:** 2024-08-13

**Authors:** Estéfani García-Ríos, Judit Pardo, Ying Su, José Manuel Guillamón

**Affiliations:** 1Food Biotechnology Department, Instituto de Agroquímica y Tecnología de Alimentos (IATA), Consejo Superior de Investigaciones Científicas (CSIC), 46980 Paterna, Valencia, Spain; judit.pardo@iata.csic.es (J.P.); guillamon@iata.csic.es (J.M.G.); 2College of Enology, Northwest A&F University, Xianyang 712100, China; yingsu@nwafu.edu.cn

**Keywords:** low temperature, *Saccharomyces*, cell growth, fermentation rate, alcoholic fermentation, nitrogen preference, wine

## Abstract

Nowadays, the wine industry carries out fermentations at low temperatures because this oenological practice clearly improves the aromatic complexity of the final wines. In addition, nitrogen content of the must also influences the quality of the wine. In this study, we carried out a phenotypic and fermentative analysis of two industrial wine *Saccharomyces cerevisiae* strains (P5 and P24) at 15 and 28 °C and three nitrogen concentrations (60, 140 and 300 mg N/L) in synthetic must. Our results show that both parameters, temperature and nitrogen, are interrelated and clearly determine the competitiveness of the wine strains and their ability to adapt at low temperatures. The best adapted strain at low temperatures decreased its competitiveness at lower nitrogen concentrations. In addition, our results show that it is not only the quantity of nitrogen transported that is important but also the quality of the nitrogen source used for wine yeast adaptation at low temperatures. The presence of some amino acids, such as arginine, branched chain amino acids, and some aromatic amino acids can improve the growth and fermentation activity of wine yeasts at low temperatures. These results allow us to better understand the basis of wine yeast adaptation to fermentation conditions, providing important information for winemakers to help them select the most appropriate yeast strain, thus reducing the economic costs associated with long and sluggish fermentations. The correlation between some amino acids and better yeast fermentation performance could be used in the future to design inactive dry yeast enriched in some of these amino acids, which could be added as a nutritional supplement during low temperature fermentations.

## 1. Introduction

Yeasts frequently face stressful environments characterized by inadequate temperatures, oxygen deficiency, medium acidity, or poor nutritional composition, such as limited nitrogen, lipids, vitamins, or mineral salts. One of their greatest challenges is coping with low nitrogen availability and fluctuating temperatures [1,2,3].

In winemaking, yeast assimilable nitrogen (YAN) in grape musts comprises a complex mix of ammonium ions and amino acids, with concentrations ranging from 60 to 2400 mg/L, depending on the grape variety and cultivation conditions [4]. Nitrogen is a vital nutrient that directly affects yeast growth [5], fermentation performance [6], and the development of sensory qualities in wine [7,8,9]. Adequate nitrogen levels are crucial for completing the fermentation process, and low nitrogen levels can lead to slow or stuck fermentations [10,11]. Yeasts can utilize different nitrogen-containing compounds, which impact growth in varying ways. These compounds are classified as either preferred (such as ammonium, glutamine, glutamate, and asparagine) or non-preferred (such as urea, proline, and allantoin) [12]. When present in a mixture, these compounds are consumed sequentially during wine fermentation. This sequential consumption depends somewhat on the availability of substrates and the specific yeast strain, likely due to the differential regulation of permeases involved in their uptake [13,14]. Preferred nitrogen sources significantly influence the growth rate during fermentation with a single nitrogen source. These preferred sources are consumed first, except for ammonium, which, although it supports optimal growth, begins to be consumed only after other preferred sources like glutamate and glutamine are exhausted [11,12].

Furthermore, temperature is a fundamental factor affecting all living organisms, which vary widely in their tolerance to different temperature ranges. As temperature fluctuates over time, most organisms display significant changes in gene expression in response to these shifts [15,16,17]. In yeast, temperature-induced changes in gene expression are primarily associated with a general environmental stress response, involving mostly transient alterations in genes related to protein synthesis, cellular growth, and metabolism [3,18,19]. Cold temperature is known to induce biochemical, biophysical, and physiological changes in cells [20]. It strengthens the interactions between the two strands of DNA and the secondary structure of mRNA, impairing transcription and translation [21,22]. Additionally, cold temperature decreases the fluidity of the lipid bilayer in membranes, increasing their rigidity and reducing transport through the cell membrane [23,24]. The rate of protein folding decreases, while conformational instability and protein denaturation increase, leading to a general reduction in enzymatic activity. The mechanisms favoring the uptake of one nitrogen source over another have been extensively investigated [12,13,25,26,27,28,29], and many studies have been done to explore the nitrogen-demanding character of *S. cerevisiae*, both phenotypically and genotypically [1,12,30,31,32]. However, few studies have explored the assimilation of nitrogen compounds when present as a complex mixture of ammonium and amino acids, as found in grape juice and at low temperature conditions (13 °C) [33]. In this previous study [33], low temperatures reduced both fermentation and growth rates in yeast cells. At 13 °C, yeast consumed less ammonium and glutamine but more tryptophan, compared to 25 °C. Additionally, low temperatures appeared to relax nitrogen catabolite repression (NCR), as shown by changes in the expression of ammonium and amino acid permeases and increased uptake of amino acids like arginine and glutamine [33].

In our work, we aimed to decipher the influence of low temperatures on nitrogen source preferences in two wine strains of *S. cerevisiae* with divergent phenotypes during low temperature fermentations [18,34]. With this aim, we evaluated the ability to grow and ferment at different nitrogen concentrations and with different sources at low temperatures (15 °C), compared with the optimum temperature (28 °C).

## 2. Materials and Methods

### 2.1. Yeast Strains and Media

We selected two industrial wine strains of *S. cerevisiae*, P5 and P24, as parent strains for their marked phenotypic growth differences at low temperatures (15 °C) [18]. P5 corresponds to the commercial name ICV-GRE strain while P24 has not been yet commercialized. Both strains were kindly provided by Lallemand Inc. (Blagnac, France).

For the competition experiments, we constructed a derivative P5 strain by replacing one copy of the *GAL1* open reading frame (ORF) with the GFP-KanMX4 deletion cassette, using the short flanking homology (SFH) method [18,35,36]. The deletion cassette was obtained using plasmid pKT127 [37] as a template. *S. cerevisiae* was transformed via the lithium acetate method [38], and transformants were selected based on their resistance to geneticin. Correct integration of the deletion cassette was confirmed by Polymerase Chain Reaction (PCR), using oligonucleotides located upstream and downstream of the cloning site.

The growth media selected for the experiments were synthetic grape must (SM), derived from that described by Quirós et al. (2013) [39], and liquid synthetic complete (SC) medium. The SM composition included 200 g/L of sugars (100 g/L glucose + 100 g/L fructose), 6 g/L of malic acid, 6 g/L of citric acid, 1.7 g/L of yeast nitrogen base (YNB) without ammonium and amino acids, anaerobic factors (0.015 g/L ergosterol, 0.005 g/L sodium oleate, and 0.5 mL/L tween 80), and 0.060 g/L of potassium disulfite. The nitrogen content and nitrogen source were modified for the different fermentations: 60 mg N/L (18 mg N/L as ammonium and 42 mg N/L in amino acid form), 140 mg N/L (42 mg N/L as ammonium and 98 mg N/L in amino acid form), and 300 mg N/L (90 mg N/L as ammonium and 210 mg N/L in amino acid form). The proportions of each amino acid in the different nitrogen concentrations were administered as previously proposed by Riou et al. (1997) [40], mimicking the proportions of these amino acids in different natural musts. The final pH of the SM was adjusted to 3.3 with NaOH. The SC was composed of 1.7 g/L yeast nitrogen base without ammonium sulfate and amino acids, 5 g/L ammonium sulfate, 20 g/L glucose, and SC drop-out (Formedium, Norfolk, UK). Inocula of the commercial yeast strains were prepared by rehydrating the dry yeasts in water, following the manufacturer’s recommendations (30 min at 37 °C). The volume required to obtain a concentration of about 2 × 10^6^ cells/mL was inoculated in the different media. Correct inoculation size was always confirmed by surface spread on a YPD rich medium (1% Yeast Extract, 2% Bacto-peptone, 2% glucose) agar plates.

### 2.2. Growth Conditions

Growth was monitored by measuring optical density at 600 nm, using a SPECTROstar Omega instrument (BMG Labtech, Offenburg, Germany). For the 28 °C experiments, measurements were taken every 30 min for 4 days after 20 s of pre-shaking. For the low-temperature assays, measurements were taken every 90 min for 5 days. Microplate wells were filled with the required volume of inoculum and 0.25 mL of the SC or SM medium to ensure an initial optical density at 600 nm (OD) of approximately 0.1, corresponding to an inoculum level of about 10^6^ cells/mL. For the nitrogen assays, the SM medium was modified with different nitrogen concentrations (0, 10, 20, 40, 60, 80, 100, 120, 140, 160, 180, and 200 mg N/L). Non-inoculated wells were included in each experimental series to determine and subtract background noise. All experiments were conducted in triplicate.

Growth parameters were calculated by fitting OD measurements vs. time to the reparametrized Gompertz equation proposed by Zwietering et al. (1990) [41]:y = D ∗ exp {−exp [((µ_max_ ∗ e)/D) ∗ (λ − t) + 1]}
where y = ln(OD_t_/OD_0_); OD_0_ is the initial OD; OD_t_ is the OD at time t; D = ln(OD_t_/OD_0_) is the asymptotic maximum; µmax is the maximum specific growth rate (1/h); and λ is the lag phase period (h) [42]. The overall yeast growth was estimated as the area under the OD vs. time curve (AUC 70 h). This parameter was calculated by integration, using the OriginPro 8.5 software (OriginLab Corp., Northampton, MA, USA).

### 2.3. Fermentation Trials

To determine the effect of nitrogen concentrations on fermentation performance, the SM was modified with a mixture of ammonium and amino acids in the following manner: 60 mg N/L (18 mg N/L as ammonium and 42 mg N/L in amino acid form), 140 mg N/L (42 mg N/L as ammonium and 98 mg N/L in amino acid form), and 300 mg N/L (90 mg N/L as ammonium and 210 mg N/L in amino acid form). The proportion of each amino acid was adjusted as previously reported by Riou et al. (1997) [40]. After the nitrogen addition, the pH of the SM was adjusted to 3.3 and filtered to sterility. Fermentations were performed at 28 (control) and 15 °C (low temperature) with continuous orbital shaking at 100 rpm. In order to avoid fluctuations in the temperature of the fermentations, the fermenters were incubated in incubators (Nüve Cooler Incubators, Ankara, Turkey) that kept the temperature constant at 28 and 15 °C, respectively. In addition, the sampling of low temperature fermentations was carried out in a refrigerated room where the temperature never exceeded 18 °C. The fermentations were carried out in laboratory-scale fermenters, using 100 mL bottles filled with 80 mL of SM. The fermentations were monitored by the density of the media (g/L), using a densitometer (Densito 30PX, Schwerzenbach, Mettler Toledo, Switzerland), and were considered complete when density reached 995 g/L. Yeast cell growth was determined by absorbance at 600 nm and by plating on a YPD medium.

### 2.4. Competition Experiments

To determine the influence of nitrogen concentrations and fermentation temperature on the yeast population in a mixed culture, competition fermentation was conducted between the *S. cerevisiae* strains P5-GFP and P24. The strains used for fermentation were first propagated in liquid YPD media overnight and then transferred to synthetic minimal (SD) media (2% glucose, 0.017% yeast nitrogen base) with 60 mg/L NH_4_Cl in order to eliminate the influence of YPD nitrogen rich media. Fermentations were carried out in 100 mL bottles with 80 mL synthetic must of three nitrogen concentrations (60, 140 and 300 mg N/L). These three nitrogen concentrations represented nitrogen-limited, nitrogen-sufficient, and nitrogen-excess fermentation conditions. All the mixed fermentations were inoculated with 10^6^ cells/mL of each strain. In order to observe the influence of temperature on competition capacity of each strain, the fermentations were carried out at 15 and 28 °C. The percentage of each strain competing throughout fermentation was monitored by replica plating from YPD to YPD-geneticin (G-418, Formedium, Norfolk, UK), as previously described [18,30]. 

### 2.5. Nitrogen Determination by HPLC

The concentration of the different amino acids and ammonium in the SM and in the different samples taken during the fermentations were analyzed on an Ultimate 3000 (Thermo Scientific, Waltham, MA, USA) high-performance liquid chromatography (HPLC) system equipped with a UV-visible detector (Thermo Scientific). The HPLC analysis method was based on Gómez-Alonso et al. (2007) [43], with some modifications. The samples were derivatized by a reacting mixture that included 430 µL of borate buffer 1 M (pH 10.2), 300 µL of methanol, 400 µL of the sample, and 12 µL of diethyl ethoxymethylenemalonate (DEEMM) in a screw-cap test tube over 30 min in an ultrasound bath. The sample was then heated at 70 °C for 2 h to allow complete degradation of excess DEEMM and reagent byproducts. The samples were diluted up to 5 times to avoid the matrix effect when measuring samples taken from different stages of fermentation [44]. Chromatography separation was performed in a C18 AccucoreTM C18 column (Thermo Scientific) at 30 °C through the binary gradient shown in Table 1. A total volume of 5 µL was injected. The composition of the solvents was as follow: phase A: ultrapure water (cleaning); mobile phase B: 100% acetonitrile; and phase C: Acetate buffer 25 mM de pH 6. The target compounds were identified according to the retention times of their corresponding standards. Quantification was performed using the calibration curves of the respective standards.

### 2.6. Extracellular Metabolites Analysis

Glucose, fructose, glycerol, ethanol and organic acids were analyzed in all the samples at the end of the fermentation process. Analytical HPLC was carried out on a Surveyor Plus Chromatograph (Thermo Fisher Scientific, Waltham, MA, USA) equipped with a refraction index detector, an autosampler, and a UV-Visible detector. Prior to injection, samples were centrifuged at 13,300 rpm for 5 min, and supernatants were filtered through 0.22 µm pore size nylon filters (Micron Analitica, Madrid, Spain) and diluted 5 or 10-folds. A total volume of 25 mL was injected into a HyperREZTM Carbohydrate Hþ 8 mm column (Thermo Fisher Scientific) assembled to its correspondent guard. The mobile phase used was 1.5 mM H_2_SO_4,_ with a flux of 0.6 mL/min and a column temperature of 50 °C. The concentration of each metabolite was calculated using external standards. Each sample was analyzed in duplicate.

### 2.7. Statistical Analysis

All of the experiments were carried out at least in triplicate. Physiological data were analyzed with the Sigma Plot 12.5 software, and the results were expressed as mean and standard deviation. To evaluate statistical significance, two tailed t-student tests were applied with a *p*-value of 0.05. Phenotypic data were fitted to the reparameterized Gompertz model by nonlinear least-squares fitting, using the Gauss-Newton algorithm as implemented in the nls function in the R statistical software v.3.0. The heatmaps were created by using the heatmap3 and complex heatmap packages in the R statistical software, v.3.0 [45].

## 3. Results

### 3.1. Effect of Temperature and Nitrogen Concentrations on Yeast Growth

In order to determine the impact of nitrogen concentrations during low temperature fermentations, yeast growth analysis was carried out. Figure 1 shows the AUC of both strains growing at 28 and 15 °C in SM with different nitrogen concentrations. As expected, increased nitrogen levels augmented the area under the curve of both strains up to a certain concentration at which the maximum value was reached and the AUC decreased. Regarding the P5 (light green) strain at 28 °C, the maximum AUC was reached at 140 mg N/L, while the maximum AUC value for the P24 (light blue) strain was reached at 60 mg N/L. P5 showed higher AUC values compared with P24 in almost all of the nitrogen concentrations up to 200 mg N/L, where their growths are almost the same. If we consider the minimum limiting concentration (CML) as the minimum concentration to reach the maximum AUC value, P24 reached the maximum AUC value at lower concentrations of nitrogen compared with P5. In other words, P24 appeared to be a less nitrogen-demanding strain than P5 at 28 °C, but P5 showed better growth in correlation with increases in nitrogen in the synthetic must.

In terms of growth at a low temperature, as expected, the AUC values were approximately half of those at 28 °C. With regard to the growth behavior of the individual strains, P5 (dark green) showed a CML of 120 mg N/L, which was close to the CML of this strain at 28 °C. For strain P24 (dark blue), the CML at a low temperature was far from the value at the optimum temperature, which was around 220 mg N/L. This strain clearly increased its growth performance at high nitrogen concentrations, with AUC values similar to the P5 strain. This means that P5 was better adapted for growth at a low temperature, as we have previously shown [17], in a range of nitrogen concentrations from 40 to 200 mg N/L, but these growth differences were equalized at higher nitrogen concentrations. These results showed that the growth performance of a yeast strain is highly dependent on the combination of these two parameters (nitrogen and temperature) in the grape must.

### 3.2. Influence of Nitrogen and Temperature on Fermentation Performance

To better understand the differences between the two strains in fermentation activity at different nitrogen concentrations and temperatures, several fermentations were carried out. The kinetics of these fermentations were estimated by calculating the time needed to ferment 5% (T5), 50% (T50), and 100% (T100) of sugars in the SM (Table 2). T5, T50, and T100 approximately matched the beginning (lag phase), middle (end of the exponential phase), and end of fermentation, respectively. Fermentations were considered complete when density reached 995 g/L. All the fermentations carried out at 28 °C were able to complete the fermentation process regardless of the nitrogen concentration. The time needed to ferment the sugars present in the must increases in the lower nitrogen concentrations in both strains, but especially in P24. Regarding the fermentations performed at a low temperature (15 °C), all the conditions needed more time to finish the process if we compare them with the same conditions at 28 °C, and this time decreased as the nitrogen concentration increased. P24 was not able to finish the fermentation at both 60 and 140 mg N/L, while at 300 mg N/L it showed a significant delay in the fermentation kinetics compared with P5. P5, which was selected for its good performance at low temperatures, needed less time to finish the fermentation process in all the conditions assayed, highlighting the cryotolerant character of this strain.

When fermentations were considered finished (density of 995 g/L) or stopped, the sugars, alcohols, and acids analysis of the final wine was carried out by HPLC. Figure 2 shows a principal component analysis (PCA) of the metabolites and the samples that allowed us to distribute the tested fermentation conditions according to the metabolic profile obtained at the end of the fermentation. The first two principal components accounted for 73% of the total variance. The topography distribution of the PCA showed that the temperature of fermentation more strongly impacted to the P24 strain (pink) because this strain was grouped according to this environmental factor and conditioned the synthesis of certain metabolites during wine fermentation. Thus, the presence of metabolites such as residual fructose and the production of citric and acetic acids were associated with strain P24 fermenting at a low temperature. Conversely, environmental factors such as temperature and nitrogen did not significantly influence the distribution of different fermentation conditions in the P5 strain (green). This lack of clear conditioning can be attributed to the strain’s cryotolerant nature, which results in a higher production of typical fermentative metabolites, including increased concentrations of ethanol, glycerol, and lactic acid. This PCA highlights the greater fermentative problems of strain P24 fermenting at a low temperature, where the wines retain a significant amount of unfermented fructose and produce acetic and citric acids, which are two classic markers of sluggish fermentations that have difficulty consuming the total amount of sugars [46,47].

### 3.3. Nitrogen Concentrations’ Influence on Fitness at Low Temperatures

In order to determine the influence of nitrogen concentrations together with temperature, competition experiments with different nitrogen concentrations were performed at both 15 and 28 °C. As the P5 strain harbored Geneticin resistance [18], the percentage of Geneticin-resistant cells in the culture, determined by replica plating on a YPD-G418 medium, was used to determine the percentage of P5 cells in the population. Figure 3 shows the percentage of each strain for every nitrogen concentration and temperature. Our results showed that both temperature and nitrogen concentrations have a significant influence on the competitiveness of the strains. At 28 °C, the P5 strain (green) took over the three mixed-culture fermentations (Figure 3D–F), with implantation percentages higher than 60% regardless of the nitrogen concentration. At 15 °C, as expected, strain P5 gradually took over the fermentation at 300 mg/L of nitrogen in cold temperatures, reaching percentages of around 80% of the total population at the end of the process (Figure 3C). However, at 140 mg/L of nitrogen, the percentage of each strain was kept at around 50% throughout the mixed fermentation when the temperature was 15 °C (Figure 3B). Conversely, when the nitrogen was lower (60 mg N/L), the P24 strain (pink) was able to dominate the mixed fermentation, highlighting the highest competitiveness of this strain in the combination of low temperature and nitrogen-limited fermentation (Figure 3A), in spite of the proved cryotolerance of P5 and the highest AUC values obtained for this condition in the above section.

### 3.4. Influence of the Concentration and Temperature in the Preferences of Nitrogen Consumption

To check if the consumption pattern of ammonium and amino acids in the different fermentations was conditioned by the nitrogen concentration and the temperature used during the process, the uptake of the different nitrogen compounds was analyzed by HPLC. Since the rate of nitrogen consumption depends on the temperature, we previously determined the time point of the low temperature fermentations (blue) at which the amount of nitrogen concentration was similar to the consumption after 24 h of fermentations at 28 °C (purple). These time points were set at 48 h for 60 mg N/L and 96 h for the 140 and 300 mg N/L conditions (Figure 4). Practically 100% of the nitrogen was consumed in the 60 mg N/L condition in both strains and at both temperatures (15 and 28 °C), and about half of the total nitrogen content in the nitrogen excess condition (300 mg N/L). Somewhat unexpected was the highest consumption of P24 at a low temperature in the 140 and 300 mg N/L conditions.

The presence of residual nitrogen, mainly in the 140 and 300 mg N/L conditions, was used to determine the preference of the individual amino acids in both strains at both temperatures. Figure 5 shows the percentage of consumed nitrogen compounds for the strains P5 and P24 at the time points analyzed in the three concentrations of nitrogen at 28 and 15 °C, respectively (blue: 75–100%; pink: 75–50%; green: 50–25%; and yellow: 25–0% of consumption). As mentioned above, most of the individual nitrogen sources were completely consumed by both strains in the nitrogen-limited condition (60 mg N/L). Only some residual concentrations of proline and cysteine were detected under these conditions. Therefore, the nitrogen-limited condition was not very informative because it was difficult to elucidate the preferences of nitrogen sources at both temperatures. The excess nitrogen condition (300 mg N/L) showed the opposite situation, in which most amino acids were not fully consumed (only lysine and methionine were 100% consumed) at both temperatures. However, in this condition and in the nitrogen-sufficient condition (140 mg/L), we can identify groups of amino acids with different preferential uptakes, with some particularities in the order of assimilation of the nitrogen substrates for each strain (Appendix A). At 28 °C, the nitrogen sources lysine, methionine, phenylalanine, tryptophan, leucine, isoleucine, arginine, threonine, aspartate, and histidine were the most rapidly consumed by the P5 strain, while the group consisting of alanine, tyrosine, glycine, and valine were the amino acids with the highest residual concentration after 24 h of fermentation. The nitrogen consumption profile of the P24 strain showed a lag with respect to the P5 strain, with higher concentrations of most amino acids after 24 h of fermentation in both 140 and 300 mg N/L conditions. Nonetheless, the order of uptake of these amino acids or preference was very similar to that observed for the P5 strain.

Regarding fermentations at 15 °C, some amino acids were also preferentially taken up to this temperature, such as lysine, leucine, methionine, phenylalanine, and isoleucine, but there were others more quickly consumed at low temperatures, such as histidine, threonine, serine, and arginine (Figure 5 and Appendix A).

However, our main aim was to identify nitrogen sources that were highly consumed at a low temperature compared to the optimum temperature. To this end, we calculated the ratio of consumption of each amino acid at 15 °C/28 °C (Figure 6). Values greater than 1 represent higher consumption at 15 °C (red). In the P5 strain, most of the amino acids were highly consumed at 28 °C, with some exceptions, such as the three branched chain amino acids (isoleucine, leucine, and valine) at 300 mg N/L. Interestingly, there were a greater number of amino acids that were preferentially consumed at a low temperature in strain P24, both at 140 and 300 mg N/L (Figure 6). This group of highly consumed amino acids at a low temperature was made up of the branched chain amino acids (isoleucine, leucine and valine), the aromatic amino acids tyrosine and phenylalanine, a mixture of the nitrogen sources considered rich or preferred, such as glutamine and ammonium, and some of the poor nitrogen sources, such as histidine, serine, and threonine [12].

### 3.5. Growth and Fermentation Analysis in Different Nitrogen Sources

In order to determine the influence of different nitrogen sources at low temperatures, we carried out a growth curve analysis of both strains in different nitrogen sources at 28 and 15 °C. Figure 7 shows the maximum specific growth rates of both strains in each of the nitrogen sources tested relative to the control (synthetic must 300 mg N/L). The bluer the color, the better the ability to grow in a particular nitrogen source compared to the control. In our experience, the best growth is supported by the mixture of amino acids and ammonium (control), so values close to 1 (white) or higher indicate a clear effect of this amino acid in improving growth. In the case of the P5 strain, most of the nitrogen sources showed a significantly worse growth than the control, with some exceptions of what are considered rich nitrogen sources (asparagine, glutamine, arginine, aspartate, and ammonium), which showed a similar growth to the control condition. However, in the case of P24, arginine was the most significant amino acid that promoted better growth at a low temperature in comparison with the control condition. Moreover, in this strain, some amino acids, not included in this group of rich nitrogen sources, promoted similar or even better growth of this strain at a low temperature. This was the case for phenylalanine, cysteine, glycine, and isoleucine. This result supports the idea that increasing these amino acids in the must could improve the growth and fermentation activity of this strain at low temperatures.

## 4. Discussion

The global wine market is currently demanding fresh and aromatic wines. A simple oenological practice to improve the aromatic quality of wines is to ferment them at low temperatures (12–15 °C), because the varietal and fermentative aromas are largely preserved [7,48]. However, this non-optimal temperature represents an additional stress to the harsh environment that alcoholic fermentation represents for wine yeasts. Our laboratory has generated new insights into the physiology, metabolism, and molecular adaptation of yeast at low temperatures, using high-throughput techniques in a global approach [18,34,49,50,51,52,53], and we have applied this new knowledge to obtain genetically improved strains through adaptive evolution [35] and interspecific hybridization [51]. In one of these studies [18], among a pool of 27 commercial yeast strains, the P5 and P24 strains were selected as candidate strains with good and poor growth behavior at low temperatures, respectively. This selection was further confirmed in another study [35], where the P5 strain was able to prevail over a pool of 27 commercial yeast strains in a long adaptive laboratory evolution (ALE) experiment at low temperatures. However, yeast growth and fermentation performance at low temperatures is strongly influenced by nitrogen concentrations [54,55]. It is known that nitrogen requirements for fermentation may be increased at low temperatures, as plasma membrane rigidity leads to less active membrane-associated permeases and also to a significant reduction in membrane transport [46,50,56]. The aim of this study was to evaluate the effect of the nitrogen content of the must on the growth and fermentation performance of an industrially well-adapted wine yeast at low temperatures.

As expected, P5 grew and fermented better at a low temperature than the P24 strain. However, these differences were remarkably dependent on nitrogen concentrations. Growth differences were minimized at high nitrogen concentrations and, conversely, it was very noticeable that the competitiveness of the P5 strain decreased at lower nitrogen concentrations in the competition experiment of both strains during fermentations at low temperatures (Figure 3). A possible explanation for the increased competitiveness of strain P24 in the combination of low temperatures and nitrogen-limited fermentation could be the higher fitness of P24 to consume nitrogen at low temperatures compared to P5 (Figure 4). In a competition experiment, a faster uptake of nitrogen compounds could be very advantageous in a situation where this transport could be affected by the rigidity of the plasma membrane, especially in a nitrogen-limited condition.

Nonetheless, it is not only the quantity of nitrogen transported that is important but also the quality of the nitrogen source used for wine yeast adaptation at low temperatures [33]. Therefore, another aim of this study was to evaluate the nitrogen preferences of both strains when fermenting at low temperatures. The consumption profile at 28 °C is very similar to that found in previous works about sequential use of nitrogen compounds by *S. cerevisiae* [13] and non-*Saccharomyces* yeasts [57] during wine fermentation. With the exception of arginine, most of these early-consumed amino acids were transported to yeast cells through the plasma membrane Ssy1-Ptr3-Ssy5 (SPS) regulated permeases, which are specifically induced in response to the presence of their substrate in the medium [13]. This nitrogen uptake profile was very similar in both strains and at both temperatures (15 and 28 °C), with some exceptions. Serine was consumed first in the low temperature fermentations (Figure 5). The uptake of this amino acid has been correlated with the maintenance of sphingolipid and phospholipid homeostasis in *S. cerevisiae* [58], something that is absolutely necessary at low temperatures, where the plasma membrane has to be reshaped to counteract the decrease in fluidity. Arginine was also preferentially taken up at low temperatures. The transport of this amino acid is mainly carried out by the permeases *CAN1* and *GAP1*, both of which are subject to the nitrogen catabolite repression (NCR) mechanism, which avoids the uptake of this amino acid while the so-called rich nitrogen sources (mainly glutamine and ammonium) are not consumed [59]. However, Beltran et al. (2006) [60] already observed a more relaxed NCR mechanism at low temperatures, as deduced from the gene expression of ammonium and amino acid permeases (*MEP2* and *GAP1*) and the uptake of some amino acids subjected to NCR (i.e., arginine). Intracellular arginine accumulation, along with other charged amino acids such as Glu and Lys, has been reported as a protective mechanism against various stresses in yeast, mainly as a cryoprotective function [61,62,63], which could explain the higher avidity of this amino acid in low temperature fermentations. Moreover, we also detected a better growth of the P24 strain at low temperatures when arginine was used as the sole nitrogen source in comparison to a complete mixture of amino acids and ammonium (control condition), as previously reported at the optimum temperature [31].

Regardless the order of uptake of amino acids, we also calculated the ratio of the total consumption at 15 and 28 °C of each amino acid and ammonium. This parameter also is very relevant to the nitrogen sources that improved growth and fermentation activity at low temperatures. The most remarkable finding was a higher consumption of the branched chain amino acids in both strains and of some aromatic amino acids in the P24 strain at low temperatures. As far as we know, tryptophan was the only amino acid previously reported to be a limiting amino acid during *S. cerevisiae* growth at low temperatures [56]; neither the other aromatic amino acids nor the branched chain amino acids were reported. The catabolization of these amino acids yields complex carbon structures (mainly higher alcohols) that can also act as cryoprotective molecules in *S. cerevisiae*. As in the case of arginine, we also detected a similar or improved growth at low temperatures when phenylalanine and isoleucine were used as the sole nitrogen sources in the SM, in spite of the fact that both of them did not promote fast growth at the optimum temperature [12].

In conclusion, the nitrogen content of grape musts is a key parameter for yeast adaptation at low temperatures, and winemakers should consider both parameters when selecting wine yeasts for these harsh fermentations. Despite the paramount importance of nitrogen quantity and quality for yeast growth and fermentation metabolism in the wine industry, we have a limited understanding of the effect of nitrogen requirements when fermenting at low temperatures. In this work, we have demonstrated how nitrogen limitation can reduce the low-temperature fitness of a well-adapted commercial strain, highlighting the urgent need to elucidate the genetic basis of this phenomenon. In addition, our results show that the presence of some amino acids can improve the growth and fermentation activity of wine yeasts at low temperatures. This is interesting information for the design of inactive dry yeast enriched in some of these amino acids, which can be added as a nutritional supplement during low temperature fermentations.

## Figures and Tables

**Figure 1 foods-13-02522-f001:**
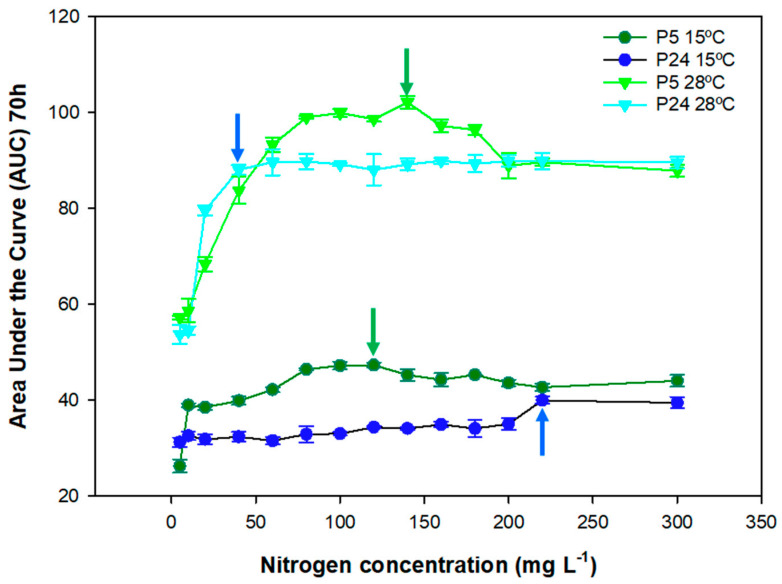
Growth analysis in SM of the yeast strains of the species *S. cerevisiae* P5 (light and deep green) and P24 (light and deep blue) represented as the area under the curve (AUC at 70 h, 1/h) as a function of the tested nitrogen concentrations, ranging from 5 to 300 mg N/L at 15 and 28 °C. Arrows indicate the minimum limiting concentration (CML) in each condition for P5 (green) and P24 (blue).

**Figure 2 foods-13-02522-f002:**
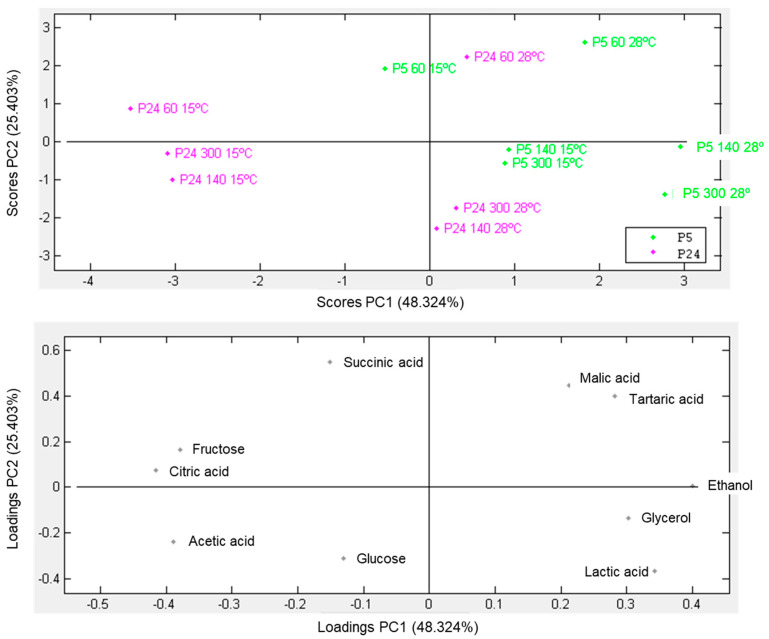
Principal component analysis (PCA) for extracellular metabolites of fermentations in SM carried out under three nitrogen conditions (60, 140, and 300 mg N/L), using P5 and P24 strains at 15 and 28 °C.

**Figure 3 foods-13-02522-f003:**
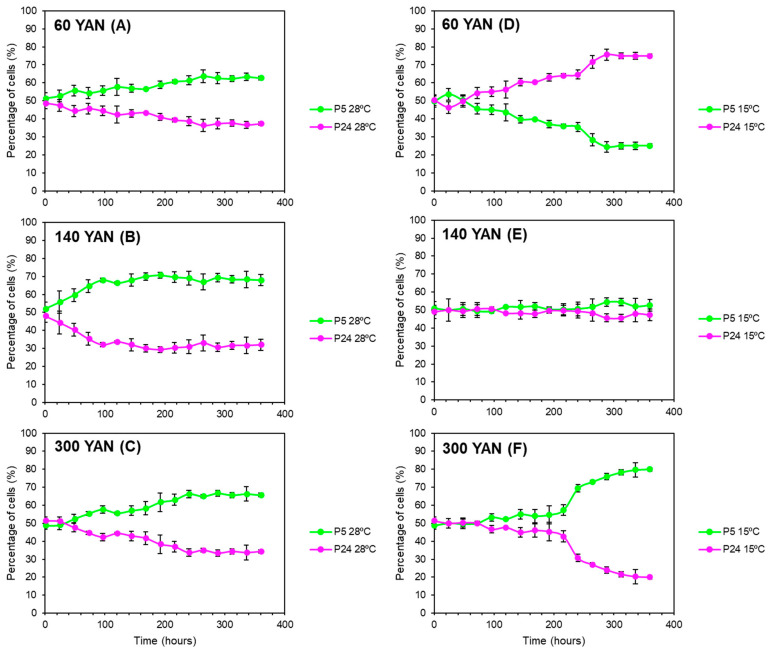
Population dynamics of a mixed culture between strains P5 (green) and P24 (pink) growing in synthetic must (SM) with three nitrogen conditions: 60 (**A**,**D**), 140 (**B**,**E**), and 300 (**C**,**F**) mg N/L at 15 (**A**–**C**) and 28 °C (**D**–**F**). The percentage of each strain was determined by replica plating in YPD + G418.

**Figure 4 foods-13-02522-f004:**
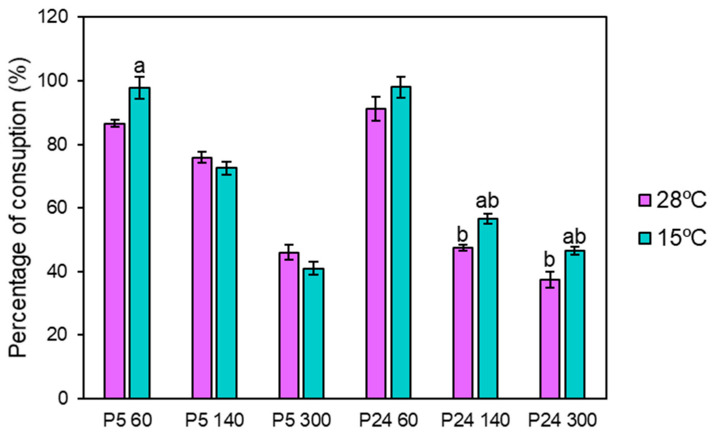
Percentage of total nitrogen consumption at three nitrogen conditions (60, 140, and 300 mg N/L) at 15 and 28 °C in SM. Since the rate of nitrogen consumption depends on the temperature, we previously determined the time point of the low temperature fermentations at which the amount of nitrogen concentration was similar to the consumption after 24 h of fermentations at 28 °C. These time points were set at 48 h for 60 mg N/L and 96 h for the 140 and 300 mg N/L conditions. In the graph, ^a^ indicates significant differences (*p* ≤ 0.05) in each strain between temperatures under the same nitrogen condition and ^b^ indicates significant differences (*p* ≤ 0.05) when comparing both strains at the same temperature.

**Figure 5 foods-13-02522-f005:**
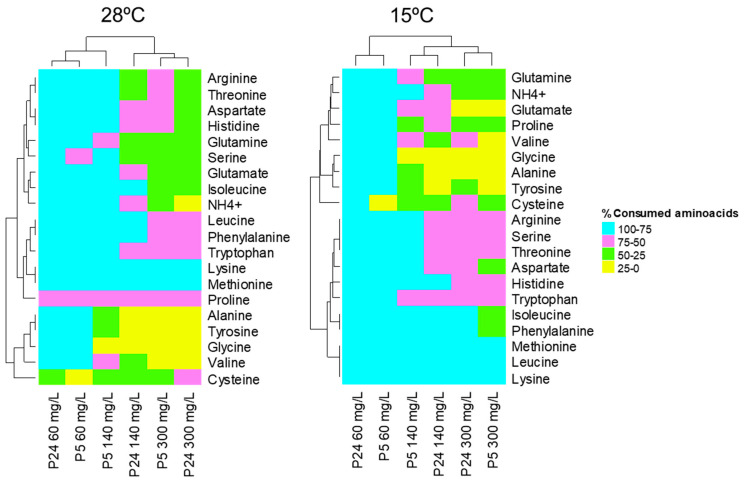
Heatmaps depicting the consumption of nitrogen compounds after 24 h of fermentation carried out by strains P5 and P24 at 28 °C for the three nitrogen conditions (60, 140, and 300 mg/L of N) and after 48 h (60 mg/L of N) and 96 h (140 and 300 mg/L of N) at 15 °C. The percentage (%) of consumption of the different nitrogen compounds present in the must is represented by colors (blue: 75–100%; pink: 75–50%; green: 50–25%; and yellow: 25–0% of consumption).

**Figure 6 foods-13-02522-f006:**
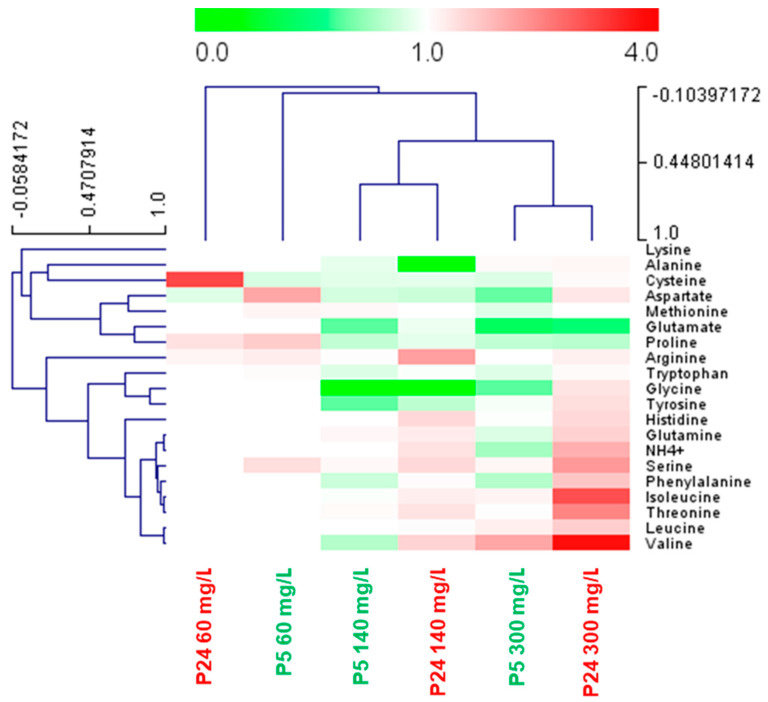
Heat map summarizing the ratio of consumption of each nitrogen compound at 15 °C/28 °C by strains P5 and P24. Fermentations were performed with three nitrogen conditions (60, 140, and 300 mg/L of N) in SM. Since the rate of nitrogen consumption depends on the temperature, we previously determined the time point of the low temperature fermentations at which the amount of nitrogen concentration was similar to the consumption after 24 h of fermentations at 28 °C. These time points were set at 48 h for 60 mg N/L and 96 h for the 140 and 300 mg N/L conditions. Red indicates higher consumption at low temperatures while green indicates higher consumption at 28 °C.

**Figure 7 foods-13-02522-f007:**
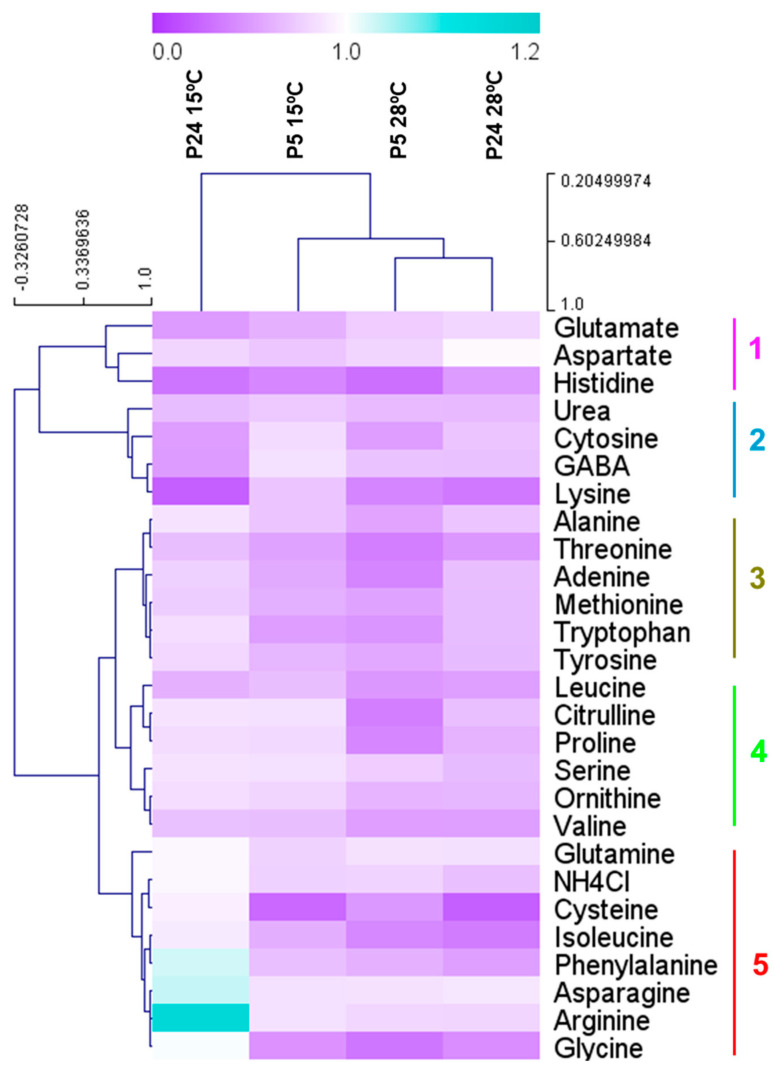
Relative maximum specific growth rate (1/h) as a function of control (200 mg N/L ammonium + amino acids) in P5 and P24 growing in pure nitrogen sources at 15 and 28 °C. Blue indicates higher growth in pure sources than in the complex mixture.

**Table 1 foods-13-02522-t001:** Eluent gradient for HPLC determination of amino acids.

Step	Time (min)	Flow Rate (mL/min)	% B	% C
Equilibration	0	1.2	5	95
Run	0	1.2	5	95
Run	3	1.2	6	94
Run	5	1.2	8	92
Run	11	1.2	10	90
Run	12.5	1.2	12	88
Run	14	1.2	18	82
Run	18	1.2	20	80
Run	21	1.2	30	70
Run	23	1.2	40	60
Run	25	1.2	75	25
Run	26	1.2	80	20
Run	30	1.2	5	95
Stop Run	35			

**Table 2 foods-13-02522-t002:** Fermentation kinetics in P5 and P24 strains at 15 and 28 °C. T5, T50, and T100 are the time (h) needed to consume the 5%, 50%, and total amount of sugars present in the must, respectively. * Significant differences (*p* ≤ 0.05) between strains at the same nitrogen concentration and temperature. # indicates a stuck fermentation before T100.

Nitrogen Concentration	Time (h)	P5 Strain		P24 Strain	
		15 °C	28 °C	15 °C	28 °C
60 mg N/L	T5	31.62 ± 4.50	12.43 ± 1.24	24.69 ± 6.66	11.50 ± 1.60
	T50	441.25 ± 70.47	70.68 ± 2.04 *	390.71 ± 8.20	130.18 ± 5.84
	T100	1333.8 ± 15.90 *	203.19 ± 5.37 *	#	368.22 ± 13.91
140 mg N/L	T5	40.35 ± 10.50	12.01 ± 1.03 *	43.03 ± 7.89	18.92 ± 0.90
	T50	193.93 ± 33.38 *	44.81 ± 1.30 *	280.13 ± 16.65	67.14 ± 2.18
	T100	585.965 ± 14.14 *	103.51 ± 4.87 *	#	161.37 ± 2.75
300 mg N/L	T5	47.08 ± 7.92	14.14 ± 4.07	34.49 ± 5.26	19.40 ± 0.14
	T50	191.31 ± 30.05 *	42.28 ± 7.30	286.99 ± 2.34	51.72 ± 2.18
	T100	452.145 ± 14.24 *	99.72 ± 5.53	1060.90 ± 9.76	109.82 ± 6.07

## Data Availability

The original contributions presented in the study are included in the article/Appendix A, further inquiries can be directed to the corresponding author.

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
