# Peer review of "Different Nitrogen Consumption Patterns in Low Temperature Fermentations in the Wine Yeast *Saccharomyces cerevisiae"

_foods, 2024, doi:10.3390/foods13162522_

Round 1

Reviewer 1 Report

Comments and Suggestions for Authors

The study effectively demonstrates the interaction between nitrogen availability and temperature on yeast growth and fermentation performance. 

While the experiments are well-designed, the manuscript lacks detailed descriptions of some controls that were used. For instance, more information on the consistency of nitrogen measurements and how variations in nitrogen content were controlled or accounted for is required. 

The discussion highlights the importance of nitrogen source quality, not just quantity. The authors could enhance the study by investigating the specific metabolic pathways involved in the assimilation of different nitrogen sources at low temperatures. Additionally, exploring how these pathways interact with other stress responses. 

Can the authors provide more details (supplementary materials) on the control conditions used in your experiments? Specifically, how did they ensure that variations in nitrogen content were accurately controlled and measured?
A vert critical point is the concern of contamination during experiments. What measures did the authors take to prevent contamination from other microorganisms during your experiments? How confident are the authors that make the results not influenced by unintended microbial activity?

Maintaining precise low-temperature conditions throughout the fermentation experiments is crucial. Any fluctuations in temperature can influence yeast metabolism and the interpretation of results. It is recommended for the authors to provide detailed descriptions of the temperature control mechanisms used and if any potential source of variability is existed. 

Some references are too old. The following references could be an alternative

1. An, H.; Li, G.; Yang, Z.; Xiong, M.; Wang, N.; Cao, X.; Yu, A. Denovo Production of Resveratrol by Engineered Rice Wine Strain Saccharomyces cerevisiae HJ08 and Its Application in Rice Wine Brewing. J. Fungi 202410, 513. https://doi.org/10.3390/jof10080513 

2. Agarbati, A.; Comitini, F.; Ciani, M.; Canonico, L. Occurrence and Persistence of Saccharomyces cerevisiae Population in Spontaneous Fermentation and the Relation with “Winery Effect”. Microorganisms 2024, 12, 1494. https://doi.org/10.3390/microorganisms12071494

Author Response

Reviewer 1

The study effectively demonstrates the interaction between nitrogen availability and temperature on yeast growth and fermentation performance.

While the experiments are well-designed, the manuscript lacks detailed descriptions of some controls that were used. For instance, more information on the consistency of nitrogen measurements and how variations in nitrogen content were controlled or accounted for is required.

Our group has a long experience in the field of nitrogen metabolism and we have set up several methods to determine nitrogen concentration and nitrogen sources in natural and synthetic musts, as well as during the fermentation process. In this case, as explained in the M&M section, the concentration of the different amino acids and ammonium were analysed by high performance liquid chromatography (HPLC). This is a reference and very accurate method for calculating both total YAN and the amount of each amino acid and ammonium. Therefore, we do not understand the concern of the reviewer regarding the nitrogen determination or the lack of controls mentioned. However, for clarity, we have added this paragraph at the beginning of the M&M section:

The concentration of the different amino acids and ammonium in the SM and in the different samples taken during the fermentations were analysed on an Ultimate 3000 (Thermo Scientific) high-performance liquid chromatography (HPLC) system equipped with a UV-visible detector (Thermo Scientific, MA). (Lines 170-173).

The discussion highlights the importance of nitrogen source quality, not just quantity. The authors could enhance the study by investigating the specific metabolic pathways involved in the assimilation of different nitrogen sources at low temperatures. Additionally, exploring how these pathways interact with other stress responses.

We agree with the reviewer that the results of this study open new avenues for a better understanding of nitrogen regulation during wine fermentation at low temperature. We are currently working to elucidate how the uptake and biosynthetic accumulation of arginine influence the tolerance to different stresses during wine fermentation such as ethanol tolerance or thermotolerance. We have identified this amino acid as a preferred amino acid at low temperature and we have also mentioned in the discussion that “Intracellular arginine accumulation, along with other charged amino acids such as Glu and Lys, has been reported as a protective mechanism against various stresses in yeast, mainly as a cryoprotective function [60–62], which could explain this higher avidity of this amino acid in low temperature fermentations”. (Lines 503-507).

Can the authors provide more details (supplementary materials) on the control conditions used in your experiments? Specifically, how did they ensure that variations in nitrogen content were accurately controlled and measured?

As mentioned above, the variations in nitrogen content were accurately controlled and measured by HPLC. Different dilutions of the same sample were injected to accurately determine the quantity of each amino acid. The specific determination of each amino acid was used to calculate the YAN (Yeast assimilable Nitrogen), expressed as mg of N/L.

A very critical point is the concern of contamination during experiments. What measures did the authors take to prevent contamination from other microorganisms during your experiments? How confident are the authors that make the results not influenced by unintended microbial activity?

We agree with the reviewer that contamination by other environmental strains is a concern when working in the context of a winery where the grape must has a high level of contamination by other yeast species and strains. However, the situation is very different in laboratory fermentations and when the matrix to be fermented is a synthetic must that has been previously sterilised. The inoculation of a strain at a high population level (2 million cells/ml) avoids the interference of possible contaminants and, moreover, the fermentation is monitored under sterile conditions. We are therefore fully confident that the inoculated strain was responsible for completing the fermentation.

We have routinely plated on YPD our experiments and used genotyping techniques such as delta elements or RFLPs of mitochondrial DNA to ensure the genetic identity of the strains. However, in our experience, this is completely unnecessary in these controlled synthetic must fermentations.

Maintaining precise low-temperature conditions throughout the fermentation experiments is crucial. Any fluctuations in temperature can influence yeast metabolism and the interpretation of results. It is recommended for the authors to provide detailed descriptions of the temperature control mechanisms used and if any potential source of variability is existed.

The fermentations were incubated in a refrigerated incubator that kept constant the temperature at 15ºC and the sampling was done in a refrigerated room that never exceed a temperature of 18ºC. We have introduced the next paragraph in M&M:

In order to avoid fluctuations in the temperature of the fermentations, the fermenters were incubated in incubators (Nüve Cooler Incubators, Ankara, Turkey) that kept the temperature constant at 28 and 15ºC respectively. In addition, the sampling of low temperature fermentations was carried out in a refrigerated room where the temperature never exceeded 18ºC. (Lines 144-148).

Some references are too old. The following references could be an alternative

  1. An, H.; Li, G.; Yang, Z.; Xiong, M.; Wang, N.; Cao, X.; Yu, A. Denovo Production of Resveratrol by Engineered Rice Wine Strain Saccharomyces cerevisiae HJ08 and Its Application in Rice Wine Brewing. J. Fungi 2024, 10, 513. https://doi.org/10.3390/jof10080513
  2. Agarbati, A.; Comitini, F.; Ciani, M.; Canonico, L. Occurrence and Persistence of Saccharomyces cerevisiae Population in Spontaneous Fermentation and the Relation with “Winery Effect”. Microorganisms 2024, 12, 1494. https://doi.org/10.3390/microorganisms12071494

We appreciate the reviewer's suggestion, but we do not find the fit of these two references with our work. However, we have tried to replace the oldest references with some more recent ones, such as:

  1. Bely, M.; Sablayrolles, J.M.; Barre, P. Description of Alcoholic Fermentation Kinetics: Its Variability and Significance. Am. J. Enol. Vitic. 1990, 41, 319–324, doi:10.5344/ajev.1990.41.4.319.

 Has been replaced by:

  1. Mendes-Ferreira, A.; Mendes-Faia, A.; Leao, C. Growth and Fermentation Patterns of Saccharomyces Cerevisiae under Different Ammonium Concentrations and Its Implications in Winemaking Industry. J. Appl. Microbiol. 2004, 97, 540–545, doi:10.1111/j.1365-2672.2004.02331.x.

  1. Lindquist, S. The Heat Shock Response. Annu. Rev. Biochem. 1986, 55, 1151–1191, doi:10.1146/annurev.bi.55.070186.005443

Has been replaced by:

  1. Verghese, J.; Abrams, J.; Wang, Y.; Morano, K.A. Biology of the Heat Shock Response and Protein Chaperones: Budding Yeast (Saccharomyces Cerevisiae) as a Model System. Microbiol. Mol. Biol. Rev. 2012, 76, 115–158, doi:10.1128/MMBR.05018-11.
  2. Morano, K.A.; Grant, C.M.; Moye-Rowley, W.S. The Response to Heat Shock and Oxidative Stress in Saccharomyces Cerevisiae. Genetics 2012, 190, 1157–1195, doi:10.1534/genetics.111.128033.

  1. Thomas, K.C.; Ingledew, W.M. Relationship of Low Lysine and High Arginine Concentrations to Efficient Ethanolic Fermentation of Wheat Mash. Can. J. Microbiol. 1992, 38, 626–634, doi:10.1139/m92-103

Has been replaced by:

  1. Gutiérrez, A.; Chiva, R.; Sancho, M.; Beltran, G.; Arroyo-López, F.N.; Guillamon, J.M. Nitrogen Requirements of Commercial Wine Yeast Strains during Fermentation of a Synthetic Grape Must. Food Microbiol. 2012, 31, 25–32, doi:10.1016/j.fm.2012.02.012.

Reviewer 2 Report

Comments and Suggestions for Authors

Dear authors, my recommendations regarding your manuscript are as follows:

1. There are used abbreviations, that are not given in words, for example PCR, SC, YPD, DEEMM, HPLC, AUC, etc.  Include an explanation for all abbreviations in the text.

2. Give citations for the used software packages, for example the OriginPro 8.5 software; the Sigma Plot 12.5 software; the R statistical software, v.3.0.

3. For clarity, insert a table for eluent gradient (rows 158-163) instead of text.

4. Precise the arrow's position for P24 28C in Figure 1.

5. Give names for all columns in Table 1.

6. Revise Figure 2. There are missing texts. Also, the word "strains'' is doubled in the explanation bellow.

7. Insert a, b, c, d … in Figure 3 for each subfigure. Give these subfigures as a reference when you explain the results.

8. A and b in text explanation under Figure 4 should not be superscripted. 

9. Where is presented Figure S1? If missing insert it.

10. Explain the meaning for different colors for all Figures in the text.

11. Precise the citations in the text, especially [64] in row 478. Almost 50% of the cited references are more than 10 years old. Where possible change them with contemporary researches.

12. The abbreviation IDY may be omitted in row 502.

Author Response

Reviewer 2

Dear authors, my recommendations regarding your manuscript are as follows:

  1. There are used abbreviations, that are not given in words, for example PCR, SC, YPD, DEEMM, HPLC, AUC, etc. Include an explanation for all abbreviations in the text.

We appreciate the reviewer's suggestion and following the recommendation we included the explanation along the manuscript.

  1. Give citations for the used software packages, for example the OriginPro 8.5 software; the Sigma Plot 12.5 software; the R statistical software, v.3.0.

We have included the citation for R statistical software.

  1. For clarity, insert a table for eluent gradient (rows 158-163) instead of text.

Following reviewer´s suggestion we have included this information in a new table (Table 1).

  1. Precise the arrow's position for P24 28C in Figure 1.

Following reviewer´s suggestion we have corrected the arrow position.

  1. Give names for all columns in Table 1.

Following reviewer´s suggestion, Table 1 (now Table 2) has been completed.

  1. Revise Figure 2. There are missing texts. Also, the word "strains'' is doubled in the explanation bellow.

The reviewer is right, we have corrected the figure.

  1. Insert a, b, c, d … in Figure 3 for each subfigure. Give these subfigures as a reference when you explain the results.

Following reviewer´s appreciation, we have now changed figure 3 accordingly.

  1. A and b in text explanation under Figure 4 should not be superscripted. 

The reviewer is right, we have corrected the figure legend.

  1. Where is presented Figure S1? If missing insert it.

Figure S1 is cited in page 10 line 378

  1. Explain the meaning for different colors for all Figures in the text.

Following reviewer´s appreciation we have improved along the manuscript the explanations (including colors) of the different figures.

  1. Precise the citations in the text, especially [64] in row 478. Almost 50% of the cited references are more than 10 years old. Where possible change them with contemporary researches.

We appreciate the reviewer's suggestion and we have tried to replace the oldest references with some more recent ones, such as:

  1. Bely, M.; Sablayrolles, J.M.; Barre, P. Description of Alcoholic Fermentation Kinetics: Its Variability and Significance. Am. J. Enol. Vitic. 1990, 41, 319–324, doi:10.5344/ajev.1990.41.4.319.

 Has been replaced by:

  1. Mendes-Ferreira, A.; Mendes-Faia, A.; Leao, C. Growth and Fermentation Patterns of Saccharomyces Cerevisiae under Different Ammonium Concentrations and Its Implications in Winemaking Industry. J. Appl. Microbiol. 2004, 97, 540–545, doi:10.1111/j.1365-2672.2004.02331.x.

  1. Lindquist, S. The Heat Shock Response. Annu. Rev. Biochem. 1986, 55, 1151–1191, doi:10.1146/annurev.bi.55.070186.005443

Has been replaced by:

  1. Verghese, J.; Abrams, J.; Wang, Y.; Morano, K.A. Biology of the Heat Shock Response and Protein Chaperones: Budding Yeast (Saccharomyces Cerevisiae) as a Model System. Microbiol. Mol. Biol. Rev. 2012, 76, 115–158, doi:10.1128/MMBR.05018-11.
  2. Morano, K.A.; Grant, C.M.; Moye-Rowley, W.S. The Response to Heat Shock and Oxidative Stress in Saccharomyces Cerevisiae. Genetics 2012, 190, 1157–1195, doi:10.1534/genetics.111.128033.

  1. Thomas, K.C.; Ingledew, W.M. Relationship of Low Lysine and High Arginine Concentrations to Efficient Ethanolic Fermentation of Wheat Mash. Can. J. Microbiol. 1992, 38, 626–634, doi:10.1139/m92-103

Has been replaced by:

  1. Gutiérrez, A.; Chiva, R.; Sancho, M.; Beltran, G.; Arroyo-López, F.N.; Guillamon, J.M. Nitrogen Requirements of Commercial Wine Yeast Strains during Fermentation of a Synthetic Grape Must. Food Microbiol. 2012, 31, 25–32, doi:10.1016/j.fm.2012.02.012

The abbreviation IDY may be omitted in row 502.

The reviewer is right, we have omitted this term

Reviewer 3 Report

Comments and Suggestions for Authors

The manuscript focuses on the impact of nitrogen consumption patterns at two temperatures 15 and 28 C in wine yeast. The research demonstrates that nitrogen quantity and quality play a crucial role in yeast growth during fermentation. Factors like temperature and nitrogen availability affect yeast competitiveness and fermentation activity, influencing wine quality. However, there is some lack of clarity concerning the low or cold temperature reported in the manuscript. While the paper seems original and valuable, it requires major revisions before final acceptance.

1.      The title " Different Nitrogen Consumption patterns at Low Temperature  in the Wine Yeast Saccharomyces cerevisiae " the title should be revised. There is some lack of clarity concerning the low temperature. Are 28°C and 15°C considered cold or low temperatures?

2.The abstract is too general and lacks details on the specific experimental conditions for strain growth, as well as the significant results obtained. Experimental conditions for strain growth and important values should be added in the abstract…..Including these specific details would enhance the clarity and comprehensiveness of the abstract, making it more informative and impactful for readers.

3.      Line 66: " However, few studies have explored the assimilation of nitrogen compounds 66 when present as a complex mixture of ammonium and amino acids, as found in grape  juice and during low temperature conditions [31] ". authors should add more details about this work, in order to introduce clearly the present research aims. Authors should make the research strategy easier for readers to understand the importance of this research

4.      Line 68:" during low temperature conditions " please correct at low temperature. Please specify the exact temperature range?

5.      Line 72: " …in cold. " Could you clarify what you mean by " cold"

6.      Line 77: " …for their marked phenotypic growth differences at low temperature…. " Could you clarify what you mean by 'low temperature' by specifying the exact temperature range?

7.      Line 80: which strain was used in this study P5 or the derivative P5 strain? This point should clarified in the methodology

8.      Line 87: " …. the experiments were synthetic grape must (SM) de-87 rived from that described by Quirós et al., (2013) [37] and SC." SC. ????? is it the synthetic must??? What is the difference between SM and SC?????

9.      Line 92 : "The assimilable nitrogen source used was administered 92 as previously proposed by Riou et al., (1997) [38]". Please explain and add more details, the reference is old??

10.  Line 93 : " Nitrogen content and nitrogen source were modified for the different fermentations as described below". Why below? Please add here the details of Nitrogen content and nitrogen source..

11.  Line 95: " …was adjusted to 3.3 with NaOH " Is the pH 3.3 optimal for the growth of the strains?

12.  Line 101 YPD??? Please  add a list of abbreviation

13.  Line 107: OD????? Add abbreviation list

14.  Line 192: AUC? Please add a list of abbreviation

15.  Line 128: Fermentations were performed at 28 and 15ºC. Are 28°C and 15°C considered cold temperatures? Generally, for many biological processes, "cold" might mean temperatures below room temperature.

16.  Line 141: On what basis did the authors select the three nitrogen concentrations (60, 140 and 300 mg N/L)?

17.  Line 151: DEEMM?? Add in a list of abbreviation

18.  Line 192: the AUC??? add a list of abbreviation

19.  Line 192 and figure 1: what is the unit of AUC?

20.  Line 219: the title of figure 1 should indicated that the strains were cultivated on synthetic grape must

21.  Line 213: " at low temperature, " see my comment about low temperature and cold…and correct, Please thoroughly review the manuscript and make the necessary correction regarding the temperature..

22.  Line 223: "…. two parameters. " indicate the two parameters clearly…

23.  Line 225-232: this should be included in the methofdology???

24.  Line 236 : at low temperature?? at 28ºC???

25.  Line 245: authors should indicate in the table the meaning of the unit of value 60, 140 and 300?? Authors should also indicate the unit o the values of T5,T50 et T100. Please make clear the table

26.  Line 245: Table 1: the title " Fermentation kinetics "??? What do the values in the table correspond to? Indicate the unit of these values???

27.  Line 261:" Conversely, the environmental factors (temperature and nitrogen) did not clearly condition the distribution of the different fermentation conditions in the P5 strain, which could be interpreted as a consequence of their cryotolerant feature, showing a higher production of typical fermentative metabolites, such as higher concentrations of ethanol, glycerol, and lactic acid."   The statement is too long and not clear, please refine it

28.  Line 268: please explain "problematic fermentations"

29.  Line 272: please correct " using strains P5 and P24 strains at 15 and 28°C. " to…  using P5 and P24 strains at …, also indicate in the title the used growth medium

30.  Line 274: the title of the sub-section" Nitrogen concentration dramatically determine the fitness at low temperature " should be changed

31.  Line 350: figure 5 showed the consumption of nitrogen for P5 and P24 cultivated separately. Could the consumption of nitrogen compounds be affected by mixed cultures of the two strains? Why did the authors not investigate nitrogen consumption in a mixed culture?

32.  Line 414: in the title" at both temperatures. " indicate the temperatures.

33.  Line 458: SPS? Abbreviation of ???

34.  Conclusion is missing. The work limits and specific recommendations or guidelines based on the study's results should be added  

Comments on the Quality of English Language

Minor editing of English language required

Author Response

Reviewer 3

The manuscript focuses on the impact of nitrogen consumption patterns at two temperatures 15 and 28 C in wine yeast. The research demonstrates that nitrogen quantity and quality play a crucial role in yeast growth during fermentation. Factors like temperature and nitrogen availability affect yeast competitiveness and fermentation activity, influencing wine quality. However, there is some lack of clarity concerning the low or cold temperature reported in the manuscript. While the paper seems original and valuable, it requires major revisions before final acceptance.

  1. The title " Different Nitrogen Consumption patterns at Low Temperature  in the Wine Yeast Saccharomyces cerevisiae " the title should be revised. There is some lack of clarity concerning the low temperature. Are 28°C and 15°C considered cold or low temperatures?

Nowadays, with the arrival of technology to accurately control the temperature of fermentations in wineries, winemakers are using this technology to shape the quality of the final product. Most red wines are fermented at 25-28ºC, which is close to the optimum yeast growth temperature. These fermentations should be completed in 5-7 days and have not been very problematic in the past. However, in the case of white and rosé wines, there is a tendency to ferment at low temperatures in order to preserve most of the varietal and fermentative aromas. These wines are fermented between 15 and 20ºC and these fermentations are clearly considered "low temperature" fermentations because the yeast has to operate very far from its optimum growth temperature. Therefore, in our study and in many other published studies on the subject, 15ºC is considered low temperature, while 28ºC should be considered the control temperature condition.

Concerning the title, we have introduced the word “fermentations” for a better clarity. The final title is “Different Nitrogen Consumption patterns at Low Temperature fermentations in the Wine Yeast Saccharomyces cerevisiae "

2.The abstract is too general and lacks details on the specific experimental conditions for strain growth, as well as the significant results obtained. Experimental conditions for strain growth and important values should be added in the abstract…..Including these specific details would enhance the clarity and comprehensiveness of the abstract, making it more informative and impactful for readers.

The abstract has been fully rewritten to increase details of the experimental conditions and significant results, according to the reviewer’s suggestions

  1. Line 66: " However, few studies have explored the assimilation of nitrogen compounds when present as a complex mixture of ammonium and amino acids, as found in grape  juice and during low temperature conditions [31] ". authors should add more details about this work, in order to introduce clearly the present research aims. Authors should make the research strategy easier for readers to understand the importance of this research

Following reviewer´s appreciation, we have now included this new paragraph in the introduction (Lines 71-76): “In this study, low temperatures reduced both fermentation and growth rates in yeast cells. At 13°C, yeast consumed less ammonium and glutamine but more tryptophan compared to 25°C. Additionally, low temperatures appeared to relax nitrogen catabolite repression (NCR), as shown by changes in the expression of ammonium and amino acid permeases and increased uptake of amino acids like arginine and glutamine.”

  1. Line 68:" during low temperature conditions " please correct at low temperature. Please specify the exact temperature range?

Following reviewer´s suggestion, we have modified the text as follows: “at low temperature (15ºC) comparing with optimum temperature (28ºC).”

  1. Line 72: " …in cold. " Could you clarify what you mean by " cold"

Following reviewer´s suggestion, we have modified the text as follows: “at low temperature (15ºC) comparing with optimum temperature (28ºC).”

  1. Line 77: " …for their marked phenotypic growth differences at low temperature…. " Could you clarify what you mean by 'low temperature' by specifying the exact temperature range?

Following reviewer´s appreciation, we have modified the text as follows: “We selected two industrial wine strains of Saccharomyces cerevisiae, P5 and P24, as parent strains for their marked phenotypic growth differences at low temperature (15ºC)”

  1. Line 80: which strain was used in this study P5 or the derivative P5 strain? This point should clarified in the methodology

We have now clarified in the methodology that derivative P5 strain was only used to perform the competition experiments.

  1. Line 87: " …. the experiments were synthetic grape must (SM) derived from that described by Quirós et al., (2013) [37] and SC." SC. ????? is it the synthetic must??? What is the difference between SM and SC?????

The reviewer is right, we have added the explanation of the term. SM is synthetic must while SC is synthetic complete media.

  1. Line 92 : "The assimilable nitrogen source used was administered as previously proposed by Riou et al., (1997) [38]". Please explain and add more details, the reference is old??

We have traditionally followed, in our studies about nitrogen metabolism during wine fermentation, the recipe of the synthetic must propose by Riou et al., (1997), mainly in terms of the proportion of the different amino acids, which attempted to reproduce the proportion of these amino acids in different the grape-musts of different grape varieties. In any case, in the previous version was not very clear and we have now modified this section in M&M (lines 102-107).

  1. Line 93 : " Nitrogen content and nitrogen source were modified for the different fermentations as described below". Why below? Please add here the details of Nitrogen content and nitrogen source.

Following reviewer´s appreciation, we have added the details of nitrogen content and nitrogen source in this section (Lines 102-107).

  1. Line 95: " …was adjusted to 3.3 with NaOH " Is the pH 3.3 optimal for the growth of the strains?

pH 3.3 is the standard value that is used in the synthetic grape must that mimics the pH of the natural grape must, which ranges between 3 to 3.6.

  1. Line 101 YPD??? Please add a list of abbreviation

The reviewer is right, we have added the explanation of the term.

  1. Line 107: OD????? Add abbreviation list

The reviewer is right, we have added the explanation of the term.

  1. Line 192: AUC? Please add a list of abbreviation

The reviewer is right, we have added the explanation of the term.

  1. Line 128: Fermentations were performed at 28 and 15ºC. Are 28°C and 15°C considered cold temperatures? Generally, for many biological processes, "cold" might mean temperatures below room temperature.

As above-mentioned, most of the red wines are fermented at 25-28ºC, which is close of the optimum yeast growth temperature. These fermentations should end up in 5-7 days and they did not use to be very problematic. However, in the case of white and rose wines, there is a trend to ferment at low temperature to preserve most of the varietal and fermentative aroma. These wines are fermented between 15 to 20ºC and, these fermentations are clearly considered “low-temperature” fermentations because yeast have to operate very far of their optimum growth temperature. Therefore, in our study, and in many other published studies of this topic, 15ºC is considered low temperature whereas 28ºC should be considered as the control condition.

  1. Line 141: On what basis did the authors select the three nitrogen concentrations (60, 140 and 300 mg N/L)?

Following reviewer´s appreciation, we have included this statement for clarification in lines 162-163: “These three nitrogen concentrations represented nitrogen-limited, nitrogen-sufficient and nitrogen-excess fermentation conditions.”

  1. Line 151: DEEMM?? Add in a list of abbreviation

The reviewer is right, we have added the explanation of the term.

  1. Line 192: the AUC??? add a list of abbreviation

The reviewer is right, we have added the explanation of the term.

  1. Line 192 and figure 1: what is the unit of AUC?

Following reviewer´s suggestion, we have included new information in Figure 1 legend. “Growth analysis in SM of the yeast strains of the species S. cerevisiae P5 (light and deep green) and P24 (light and deep blue) represented as the area under the curve (AUC at 70h, 1/h) as a function of the tested nitrogen concentrations, ranging from 5 to 300 mg N/L at 15 and 28°C. Arrows indicate the minimum limiting concentration (CML) in each condition for P5 (green) and P25 (blue).”

  1. Line 219: the title of figure 1 should indicated that the strains were cultivated on synthetic grape must

Following reviewer´s suggestion, we have included new information in Figure 1 legend. “Growth analysis in SM of the yeast strains of the species S. cerevisiae P5 (light and deep green) and P24 (light and deep blue) represented as the area under the curve (AUC at 70h, 1/h) as a function of the tested nitrogen concentrations, ranging from 5 to 300 mg N/L at 15 and 28°C. Arrows indicate the minimum limiting concentration (CML) in each condition for P5 (green) and P25 (blue).”

  1. Line 213: " at low temperature, " see my comment about low temperature and cold…and correct, Please thoroughly review the manuscript and make the necessary correction regarding the temperature.

Following reviewer´s suggestion, we have now clarified this point along the manuscript.

  1. Line 223: "…. two parameters. " indicate the two parameters clearly…

Following reviewer´s suggestion, we have clarified this point: “These results showed that the growth performance of a grape must is highly dependent on the combination of these two parameters (nitrogen and temperature).”

  1. Line 225-232: this should be included in the methodology???

This information has been included in the methodology section.

  1. Line 236 : at low temperature?? at 28ºC???

Following reviewer´s suggestion, we have clarified this point: “Regarding the fermentations performed at low temperature (15ºC), all the conditions needed more time to finish the process if we compare with the same condition at 28ºC, and this time decreased as the nitrogen concentration increased.” (lines 258-261)

  1. Line 245: authors should indicate in the table the meaning of the unit of value 60, 140 and 300?? Authors should also indicate the unit o the values of T5,T50 et T100. Please make clear the table

Following reviewer´s suggestion, Table 1 (now Table 2) has been completed.

  1. Line 245: Table 1: the title " Fermentation kinetics "??? What do the values in the table correspond to? Indicate the unit of these values???

Following reviewer´s suggestion, Table 1 (now Table 2) has been completed.

  1. Line 261:" Conversely, the environmental factors (temperature and nitrogen) did not clearly condition the distribution of the different fermentation conditions in the P5 strain, which could be interpreted as a consequence of their cryotolerant feature, showing a higher production of typical fermentative metabolites, such as higher concentrations of ethanol, glycerol, and lactic acid."   The statement is too long and not clear, please refine it

Following reviewer´s suggestion, we have refined this statement: “Conversely, environmental factors such as temperature and nitrogen did not significantly influence the distribution of different fermentation conditions in the P5 strain (green). This lack of clear conditioning can be attributed to the strain's cryotolerant nature, which results in a higher production of typical fermentative metabolites, including increased concentrations of ethanol, glycerol, and lactic acid.” (Lines 293-297)

  1. Line 268: please explain "problematic fermentations"

Following reviewer´s suggestion, we have included this explanation “fermentations with problems to consume the total amount of sugars”. (Lines 300-301)

  1. Line 272: please correct " using strains P5 and P24 strains at 15 and 28°C. " to…  using P5 and P24 strains at …, also indicate in the title the used growth medium

Following reviewer´s suggestion, the legend of Figure 2 has been modified accordingly: “Figure 2. Principal component analysis (PCA) for extracellular metabolites of fermentations in SM carried out under three nitrogen conditions (60, 140, and 300 mg N/L) using P5 and P24 strains at 15 and 28°C.”

  1. Line 274: the title of the sub-section"Nitrogen concentration dramatically determine the fitness at low temperature " should be changed

The title of this subsection has been modified: “Nitrogen concentration influence on the fitness at low temperature”

  1. Line 350: figure 5 showed the consumption of nitrogen for P5 and P24 cultivated separately. Could the consumption of nitrogen compounds be affected by mixed cultures of the two strains? Why did the authors not investigate nitrogen consumption in a mixed culture?

We appreciate the reviewer's suggestion, as this experiment would have been very interesting. However, we feel that it is outside the scope of our study. We studied the differences in nitrogen consumption in two strains with different adaptations during low-temperature fermentations. The study of consumption in a mixed culture would have shed light on the microbial interactions when two strains compete for the same resources. As mentioned above, this type of study is interesting but different in scope from the present study.

  1. Line 414: in the title" at both temperatures. " indicate the temperatures.

The reviewer is right, we have added the explanation of the term.

  1. Line 458: SPS? Abbreviation of ???

The reviewer is right, we have added the explanation of the term.

  1. Conclusion is missing. The work limits and specific recommendations or guidelines based on the study's results should be added  

The last paragraph of the Discussion is indeed the conclusions of the study, but we have not separated in a different section because, according to the Journal instructions, it is not mandatory a “Conclusions” section.

Round 2

Reviewer 1 Report

Comments and Suggestions for Authors

The authors have responded to the comments accordingly